# Study on the Influence of Microinjection Molding Processing Parameters on Replication Quality of Polylactic Acid Microneedle Array Product

**DOI:** 10.3390/polym15051199

**Published:** 2023-02-27

**Authors:** Wenqian Yu, Junfeng Gu, Zheng Li, Shilun Ruan, Biaosong Chen, Changyu Shen, Ly James Lee, Xinyu Wang

**Affiliations:** 1Department of Engineering Mechanics, Dalian University of Technology, Dalian 116024, China; 2Ningbo Research Institute of Dalian University of Technology, Ningbo 315016, China; 3The William G. Lowrie Department of Chemical and Biomolecular Engineering, The Ohio State University, Columbus, OH 43210, USA

**Keywords:** microneedles, microinjection molding, processing parameters, microinjection mold design, PLA

## Abstract

Biodegradable microneedles with a drug delivery channel have enormous potential for consumers, including use in chronic disease, vaccines, and beauty applications, due to being painless and scarless. This study designed a microinjection mold to fabricate a biodegradable polylactic acid (PLA) in-plane microneedle array product. In order to ensure that the microcavities could be well filled before production, the influences of the processing parameters on the filling fraction were investigated. The results indicated that the PLA microneedle can be filled under fast filling, higher melt temperature, higher mold temperature, and higher packing pressure, although the dimensions of the microcavities were much smaller than the base portion. We also observed that the side microcavities filled better than the central ones under certain processing parameters. However, this does not mean that the side microcavities filled better than the central ones. The central microcavity was filled when the side microcavities were not, under certain conditions in this study. The final filling fraction was determined by the combination of all parameters, according to the analysis of a 16 orthogonal latin hypercube sampling analysis. This analysis also showed the distribution in any two-parameter space as to whether the product was filled entirely or not. Finally, the microneedle array product was fabricated according to the investigation in this study.

## 1. Introduction

Microneedles are regarded as a promising modern painless transdermal therapy technology. Microneedles can be used for drug delivery, blood extraction, and even beauty applications, without pain or scarring [1,2]. There are various kinds of microneedles, including microneedles made of different materials, such as silicon [3,4,5], metals [6,7], and polymers [8,9,10], solid or hollow microneedles [11], dissolvable [12] or biodegradable microneedles, etc.

The fabrication method is highly dependent on the materials and the dimensions of the microneedles, especially for polymer materials. For instance, Suzanne et al. [13] made a safe metal microneedle array by using stainless steel wire, with a diameter of 200 μm. Hyunjoo et al. [3] fabricated a silicon microneedle with an embedded microchannel. Using this microneedle, they achieved the local delivery of a dye to a target region, during a deep brain drug infusion study. Natalia et al. [4] fabricated a silicon microneedle by photolithography. There were multi-reservoirs and electrodes on this silicon microneedle, to monitor the in vivo neurochemical behavior. Le et al. [14] used a silicon mold to fabricate the rapidly separable poly(lactic-co-glycolic acid) (PLGA)/polylactic acid (PLA) microneedle patches, for the delivery of levonorgestrel (LNG) for emergency contraception. Wang et al. [11] used a polyvinyl alcohol (PVA) sacrificial template to make poly(methyl methacrylate) (PMMA) nanonozzle arrays (5 μm in height and 50–100 nm in diameter).

Micro-injection molding (μIM) is a cost-effective technology for the fabrication of thermoplastic products with microfeatures, such as biofluidic systems [15,16,17], and patches for the influenza vaccine [18] and cancer vaccine [19]. The precision of replication of the microfeatures is the key quality for a micro-injection molding product. There are many studies that focus on the precision of replication, by investigating the influence of process parameters, including melt temperature, mold temperature, injection velocity, and holding pressure, etc. For instance, Zhou et al. [20] studied the influence of mold temperature and cooling time on the filling fraction. A higher mold temperature was suggested, since it can improve the filling fraction. Higher packing pressures and packing times should also be employed, because higher parameters in the post-filling stage can cause larger residual stresses. Lucchetta et al. [21] and Liou et al. [22] revealed that higher temperatures, pressures, and injection speeds could effectively improve the filling fraction of microfeatures. Yu et al. [8] conducted a study on the effect of the injection velocity on the filling behaviors of microcavities. Masato et al. [23] studied the effects of mold temperature, holding pressure, and injection velocity on improving the filling fraction of micropillars.

On the other hand, there are also some studies that have contributed to solving the replication issue of the microfeatures, by designing scientific molds [24]. One issue is trapped air, which prevents the flow of the melt to fill the microcavity. Marco et al. [25] made a venting path at the end of the filling direction, combined the cavity air evacuation system on the mold, and investigated the filling length when varying different processing parameters, such as injection speed and mold temperature. The results indicated that a venting system should be used at higher mold temperatures, otherwise the mold temperature will be reduced by convection, due to evacuation. Demolding is also worthy of note, for improving the replication quality when the mold temperature is around the glass transition temperature (Tg). There will be a higher adhesive or frictional force between the polymer and the mold materials, which can destroy the microfeatures during ejection, and reduce the replication quality. Maria et al. [26] showed that inefficient demolding of the nanostructures limits the attainable replication quality at higher mold temperatures.

Simulation is a significant method to evaluate the replication quality of μIM products. There are sufficient theories to support the reliability of the simulation results. Currently, there is much available commercial simulation software (e.g., Autodesk Moldflow Insight^®^, Modex 3D^®^, Sigmasoft^®^, Accelrys Materials Studio^®^, etc.) which can provide powerful simulation functions (filling [27], warpage [28,29], shrinkage [30,31], etc.) to evaluate the design of the processing parameters, the product, and the mold system at an early stage, which can shorten the development time.

In this study, a PLA in-plane microneedle array product was fabricated by μIM. The product and the mold were designed. Prior to the actual fabrication, we needed to solve the replication issue. We focused on the filling behavior, by considering the effect of different processing parameters, including injection time, melt temperature, mold temperature, and packing pressure. Shrinkage and warpage analyses were not performed in this study. The filling fraction of the product was simulated using Autodesk Moldflow Insight 2018. Based on the analysis in this study, the in-plane microneedle array product was finally fabricated.

## 2. Product and Geometry

The scheme of our in-plane microneedle array product is illustrated in Figure 1. There are five microneedles connected to the base portion. The structure of a single microneedle is taken from [32]. One difference, is that our product can be connected to the injection system at the end of the base portion, such as a micropump or syringe, which can deliver drugs periodically and continuously. The other difference is that there are two trapezoidal portions in our product, the solid portion (400 μm), in front of the microneedle, and the main microchannel portion (600 μm). This design is to reduce the resistance during insertion into the dermis, and to enhance the stiffness of the microneedle. The length, thickness, and width characteristics of a single microneedle are 1.2 mm, 150 μm, and 300 μm, respectively. There is a T-shaped microchannel on the microneedle. The length, width, and depth of the T-shaped microchannel are 800 μm, 100 μm, and 50 μm, respectively. The width of the top is 100 μm. There is a lobby (main wide channel) on the base portion, in order to provide sufficient flux. The base portion can be sealed by a thin polymer protector. We can also bond two pieces together by ultrasonic welding. Then, we can obtain a 300 μm thick microneedle array product. The product can be connected to any drug delivery system, as mentioned above.

The runner system is a simple one, as illustrated in Figure 1. The length of the sprue is 50 mm. The diameters at the beginning and the end of the sprue are 3 mm and 8 mm, respectively. The diameter of the main circular runner is 8 mm, while the diameter of the second level runner is 6 mm. The lengths of both the main runner and second level runner are 40 mm. In order to reduce the time consumption of the simulation, two kinds of mesh were used in this study. A beam element was employed for the whole runner system, including the sprue. There are 34 beam elements and 35 nodes. A 3D mesh of tetrahedral elements was employed for the microneedle part. There are 2,231,732 3D tetrahedral elements, with 5 element layers in the thickness direction, in this study, to ensure the precise simulation of the filling fraction. There are 468,620 nodes in the mesh. The 3D mesh was generated by the Hypermesh software, and the simulation was performed by Autodesk Moldflow Insight 2018.

## 3. Materials and Properties

PLA is a well-known biodegradable polymer material, which has been proven to have very good biocompatibility with humans. In this paper, we used PLA, from NatureWorks (trade 3251D, USA), to investigate the filling behavior of the microneedle. The melt density and solid density of PLA 3251D are 1.05 and 1.20 g/cm^3^, respectively. As a reference in this study, the recommended mold temperature range from the Moldflow database is [4 °C, 40 °C]. The recommended melt temperature range is [160 °C, 230 °C]. The constitutive equation of the polymer melt is regarded as a non-Newtonian fluid in many commercial injection molding software programs, which follows the Newtonian equation as:(1)$$\tau_{ij}=\eta \left(T,p,\dot{\gamma }\right)\cdot {\dot{\gamma }}_{ij}$$
where ${\dot{\gamma }}_{ij}$ is the rate of strain tensor. The viscosity of the melt, $\eta \left(T,p,\dot{\gamma }\right)$, is a highly non-linear function of temperature *T*, pressure *p*, and shear rate $\dot{\gamma }$.$\dot{\gamma }$ is defined as:(2)$$\dot{\gamma }=\sqrt{\frac{1}{2}\sum_{i=1}^{3}\sum_{j=1}^{3}{\dot{\gamma }}_{ij}{\dot{\gamma }}_{ji}}$$

Many formulas of viscosity have been used to describe the shear thinning behavior. The Cross-WLF viscosity model was employed in this study, as:(3)$$\eta =\frac{\eta_{0}}{1+{\left(\frac{\eta_{0}\dot{\gamma }}{\tau^{*}}\right)}^{1-n}}$$
where,
(4)$$\eta_{0}{=D}_{1}\exp\left\{-\frac{A_{1}\left(T-T*\right)}{A_{2}+\left(T-T*\right)}\right\}$$
(5)$$T*=D_{2}+D_{3}\cdot p$$
(6)$$A_{2}={\bar{A}}_{2}+D_{3}\cdot p$$
where *τ**, *n*, $D_{1}$, $D_{2}$, $D_{3}$, $A_{1}$, and ${\bar{A}}_{2}$ are constants, which are listed in Table 1 for PLA 3251D. The viscosity curves are given in Figure 2a.

Moreover, the material state varies during flow. The melts will change into solids when the temperature is under the transition temperature. We focused particularly on the filling fraction of the microneedle at different temperature and pressure conditions. Therefore, a precise model could powerfully support the simulation work. The modified 2-domain Tait pvT model was used in this study, which is:(7)$$\upsilon \left(T,\;p\right)=\upsilon_{0}\left(T\right)\left[1-C\ln\left(1+\frac{p}{B\left(T\right)}\right)\right]+\upsilon_{t}\left(T,p\right)$$
where,
$$\upsilon_{0}\left(T\right)=\{\begin{array}{c} b_{1m}+b_{2m}\left(T-b_{5}\right)\;\;\;\;\;if\;T>T_{t}\\ b_{1s}+b_{2s}\left(T-b_{5}\right)\;\;\;\;\;\;if\;T<T_{t} \end{array}\;B\left(T\right)=\{\begin{array}{c} b_{3m}\exp\left[-b_{4m}\left(T-b_{5}\right)\right]\;\;\;\;\;if\;T>T_{t}\\ b_{3s}\exp\left[-b_{4s}\left(T-b_{5}\right)\right]\;\;\;\;\;\;if\;T<T_{t} \end{array}\;\upsilon_{t}\left(T,p\right)=\{\begin{array}{l} 0\;\;\;\;\;\;\;\;\;\;\;\;\;\;\;\;\;\;\;\;\;\;\;\;\;\;\;\;\;\;\;\;\;\;\;\;\;if\;T>T_{t}\\ b_{7}\exp\left[b_{8}\left(T-b_{5}\right)-b_{9}p\right]\;\;\;\;\;if\;T<T_{t} \end{array}\;T_{t}\left(p\right)=b_{5}+b_{6}\cdot p$$
where *C* = 0.0894, *b*_1_~*b*_9_ are material constants, which are listed in Table 2. $T_{t}$ is the transition temperature, 105 °C for PLA 3251D. The pvT curves are plotted in Figure 2b. 

The parameters in Table 1 and Table 2 are from the database in the Moldflow software. Figure 2 was plotted according to these parameters. All parameters were measured by the Autodesk Moldflow company.

## 4. Results

### 4.1. Influence of Injection Time

The first injection parameter that we studied was injection time. Polymer melt filled the cavity at the given injection time. Filling fraction was used as the key objective, which was defined as the ratio between the portion that was filled by polymer material and the whole volume of the cavity. In this section, we chose five cases, which were, 0.5 s, 1.0 s, 2.0 s, 4.0 s, and 6.0 s as injection times, and kept melt temperature (190 °C), mold temperature (25 °C), packing pressure (80% of the final injection pressure), and packing time (10 s) the same for all cases. The parameters and simulated filling fractions are listed in Table 3. The relationship between injection time and filling fraction is plotted in Figure 3. The results show that the filling fraction decreased as the injection time increased. The microneedle array was filled only if the injection time was 0.5 s, at other prescribed processing parameters. Short shot appeared as the injection time increased. The case with the filling fraction of 6.0 s was the worst condition. Only 66.21% of the cavity was filled by the polymer material.

The filling contours at the end of filling for each case are shown in Figure 4, including the entire microneedle array product and local view of the microneedle array. The base portion was fully filled for 0.5 s, 1.0 s, and 2.0 s. However, the side of the base portion was not filled sufficiently from 4.0 s, although the lobby portion was fully filled. The most severe example of this happened when the filling time was 6.0 s. Here, even the lobby portion was not filled sufficiently. The filling difference in the product was very small in the 0.5 s case, since the filling was fast. This filling difference increased as the injection time increased. The microneedle array portion was fully filled in the 0.5 s case. The filling difference was small as well. The middle microneedle was not filled, while the other four microneedles were sufficiently filled in the 1.0 s case. There was a tiny cavity remaining at the end of this microneedle. However, all the microneedles were not sufficiently filled by polymer material at filling times of 2.0 s and above. It is worth noting, that the filling fraction of the middle microneedle was the smallest for the 1.0 s, 2.0 s, and 4.0 s cases, while it was the largest for the 6.0 s case. This interesting filling phenomenon will be discussed in the following discussion section.

### 4.2. Influence of Melt Temperature

The second injection parameter was the melt temperature. In this section, we chose four cases, which were 190 °C, 200 °C, 220 °C, and 230 °C, and kept the injection time (2.0 s), mold temperature (25 °C), packing pressure (80% of the final injection pressure), and packing time (10 s) the same for all cases. The parameters, and simulated filling fractions, are listed in Table 4. The relationship between injection time and filling fraction is plotted in Figure 5. There is an ascending trend in Figure 5, as the melt temperature increased. The products were entirely filled (100%) in the cases where the melt temperatures were 220 °C and 230 °C. Short shots appeared in the 190 °C and 200 °C cases, but filling was still higher than 95%. 

The filling contours at the end of filling, for each case, are shown in Figure 6. The base portion was fully filled for all cases. The microneedle portions in the 190 °C and 200 °C cases were not filled well. The filled fraction of the central microneedle was the worst in both the 190 °C and 200 °C cases. The microneedle array was almost filled in the 200 °C case.

### 4.3. Influence of Mold Temperature

The third injection parameter was the mold temperature. In this section, we chose four cases, which were 20 °C, 25 °C, 30 °C, and 40 °C, and kept injection time (2.0 s), melt temperature (190 °C), packing pressure (80% of the final injection pressure), and packing time (10 s) the same for all cases. The parameters and simulated filling fractions are listed in Table 5. The relationship between injection time and filling fraction is plotted in Figure 7. The results show that the filling fraction increased as the mold temperature increased. The microneedle array was not entirely filled in any of the cases. Short shot happened in all cases. The filling fraction of the 20 °C case was the worst, only 97.27%. The filling fraction did not appear to change (97.70%) when the mold temperature was increased by 5 °C. The filling fraction in the 40 °C case was the best, at 99.72%.

The filling contours at the end of filling, for each case, are shown in Figure 8, including the entire microneedle array product and local view of the microneedle array. The base portion was fully filled for all cases. However, short shot happened for the microneedle array portion for all cases. The total fill time decreased by 10.35%, from 2.396 s to 2.148 s, as the mold temperature increased from 20 °C to 40 °C. The filling fraction increased by 2.46% accordingly.

### 4.4. Influence of Packing Pressure

In this section, the results of the influence of packing pressure on the filling fraction are summarized. We chose four cases, which were 60%, 80%, 90%, and 100% of the final injection pressure, and kept injection time (2.0 s), melt temperature (190 °C), mold temperature (25 °C), and packing time (10 s) the same for all cases. The parameters and simulated filling fractions are listed in Table 6. The relationship between injection time and filling fraction is plotted in Figure 9. The results show that the filling fraction increased as the packing pressure increased. The microneedle array was only entirely filled for the case of 100% packing pressure. Short shot happened for other cases. The filling fraction of the 60% case was the worst under the given conditions, with a filling fraction of only 81.57%.

The filling contours at the end of filling for each case are shown in Figure 10. The base portions of the 60% and 80% cases were not filled by the polymer. In the 90% case, the base portion was filled but the microneedle portion was not. The filling fraction of the microneedle was improved as the packing pressure increased, for both the central needle and the side needles.

### 4.5. Influence of Packing Time 

Packing time should be another parameter that could improve the filling fraction, since melts can be pushed into the cavity under packing pressure. In this section, we also investigated the effect of the packing time. We chose three different packing times in the post-filling stage, 1 s, 3 s, and 5 s. The purpose was to show the effect on the filling fraction under different packing times. The parameters are tabulated in Table 7. The injection time was 2.0 s. The melt temperature and mold temperature were 190 °C and 20 °C, respectively. The packing pressure was 100% of the final injecting pressure. However, the results showed that there was no apparent difference between these cases. The filling fraction did not change after 3 s packing. The filling contours are shown in Figure 11. The side microneedles were filled well, while the inner microneedles were similar between different packing times, shown in Figure 11. This indicates that the packing time did not contribute much during the post-filling stage in this study.

## 5. Discussion

For our microneedle array part, the melts first entered the cavity of the base portion and hit the wall of the microcavity array. The flux is highly related to the dimension of the cross-sectional area of the cavity. The flux varies as first power of pressure drop down to the cavity and two to power of the cross-sectional area of the cavity. The characteristic cross-sectional area of the single microneedle cavity was 0.045 mm^2^, while the characteristic cross-sectional area of the base portion of the cavity was 15 mm^2^. Therefore, the majority of the melts first ran forward to fill the remaining cavity of the base portion. Since the flow resistance was much higher than the cavity of the base portion, only a small amount of the melts tried to fill the cavity of the microneedle portion under pressure.

Many factors can influence the filling of the cavity. Injection time is one of the most important processing parameters. A very short injection time will cause a high flow rate and high injection pressure. This can decrease the viscosity of the melts, according to Equation (1). At this condition, melts filled the cavity rapidly, with high injection pressure and lower viscosity. However, the melts ran relatively more slowly as the injection time was increased; more heat was taken away from the mold interface to the coolant. The viscosity of the melts increased accordingly. This was not good for filling the microneedle array cavity, and resulted in short shot, due to the small dimensions of the microneedle array cavity. The results of the injection time study showed a decreasing trend as the injection time went up (Figure 4). Even the base portion was not filled well at a longer injection time. With injection times of 1.0 s, 2.0 s, and 4.0 s, the central microneedle was not filled, while for other injection times the central microneedle was entirely filled. Similar phenomena happened in other conditions, for instance in the 200 °C melt temperature, 30 °C mold temperature case. The reason was that the melt of higher temperature ran through the base cavity. It was hard to inject melts through the microneedle cavity due to flow resistance. Then, the temperature of the melt, which had slowly entered the microneedle cavity, dropped during cooling. The material changed state from a melt into a solid, according to the modified 2-domain Tait pvT model (Equation (7)). However, the material was still in the melt state at the front of the flow. The temperature of the melt in the side microneedle cavities was higher than the central microneedles. The side cavities were eventually filled better than the central microneedles. For some cases, there was not enough opportunity to fill the side microneedle cavities since the melt had already solidified. In that case, the height of the side microneedle was lower than the central one. We have observed similar filling phenomena in our previous study [8], which was about micropillar array products. In that study, the micropillar arrays were perpendicular to the base surface, which were another kind of product but showed similar phenomena.

Since the viscosity of a polymer material decreases at higher temperatures, the pressure required to fill the cavity, and the shear stress, are relatively low. Melts will fill the cavity more easily at higher temperatures, especially for the microneedle array portion. The mold temperature has a similar effect as melt temperature, in terms of the shear stress and pressure. However, the difference of the mechanism is that the effect of the mold temperature starts from the interface between the cavity surface and the polymer material, and progresses in the thickness direction. The heat exchange at higher mold temperatures is smaller than at lower mold temperatures. The temperature of the melt at higher mold temperatures, drops more slowly than is the case at lower mold temperatures. Figure 12 shows the filling state at four different mold temperatures. The injection time was 2.0 s. The melt temperature was 190 °C. The packing pressure was 80% of the final injecting pressure. The entire product cavity was filled more at a higher mold temperature, including the base portion and the microneedle array portion.

The packing process is the post-filling stage. In this study, the packing pressure was a fraction of the final filling pressure. The cavity was filled under the same flow rate as during the filling stage. If the cavity was not filled entirely, the melts were still injected into the cavity under the constant packing pressure during the post-filling stage. A higher pressure has a greater ability to feed in melts than a lower pressure. Figure 13 shows the filling state under different packing pressures. The injection time was 4.0 s. The melt temperature and mold temperature were 190 °C and 25 °C, respectively. The cavity was filled more with melts under higher packing pressures during the post-filling stage, for both the base portion and microneedle array portion. 

Moreover, we also studied the influence of injection time, melt temperature, mold temperature, and packing pressure, by using the orthogonal latin hypercube sampling (OLHS) method [33]. Sixteen injection molding parameters (16 levels, L16) were determined by OLHS, in the ranges shown in Table 8, which are tabulated in Table 9. There were five levels that reached 100% filling state, which were ID1, 3, 4, 7, and 8. These results are plotted in Figure 14. All the samples which were fully filled are shown by star symbols, while the worst sample is represented by a red square. This shows the distribution of all the samples in any two-parameter space, and further shows that the filling fraction was a combination of all injection molding parameters. For instance, a longer injection time should cause a smaller filling fraction. However, there is a sample in the subfigure of injection time, under which the filling fraction was not the worst, since other parameters affected the filling process.

If we compare with other cases, ID1, ID3, ID4, and ID8, the injection time, melt temperature, and mold temperature of ID7 were lower than in the other cases, except the packing pressure. A lower injection time means a high fabrication efficiency, and a lower temperature means lower energy consumption. Overall, it should be a good choice of low cost and high efficiency compared to the others. At the end of this study, we made the microinjection mold shown in Figure 15a, and employed the parameters of ID 7 in Table 7. A German Boy 22A injection molding machine was used to fabricate the product. It is very hard to define whether the part is 100% filled or not during actual fabrication. In this study, we focused on the volume filled by polymer, especially for the microfeature portion. We used a microscope to check whether the microfeature was filled entirely or not. The microneedle array product was eventually fabricated successfully, as shown in Figure 15b,c.

## 6. Conclusions

In this study, we designed an in-plane microneedle array product. In order to fabricate the product successfully, the influences of different injection molding parameters were investigated. The results showed that the injection time, melt temperature, and packing pressure affect the filling fraction of the microneedle product, while mold temperature and packing time do not, especially packing time. It was hard to fill the microcavities if the temperature of the melts was low, either in the filling stage or the post-filling stage. For most cases, the base portion was filled with polymer material. However, the microcavities were not. The results also showed that, if the base portion was filled before the packing stage, the side microcavities filled more than the central one. This was because the temperature of the melts in the side microcavities were still higher than the melts in the central microcavity. Therefore, the melts in the side microcavities had a relatively lower flow resistance than in the central one. We eventually made the mold, and fabricated the microneedle array product based on the analysis of the influence of the processing parameters. We are also designing a new mold with a larger aspect ratio. Therefore, the contribution of the work in this study will be a good guide for the design and fabrication of the new product.

## Figures and Tables

**Figure 1 polymers-15-01199-f001:**
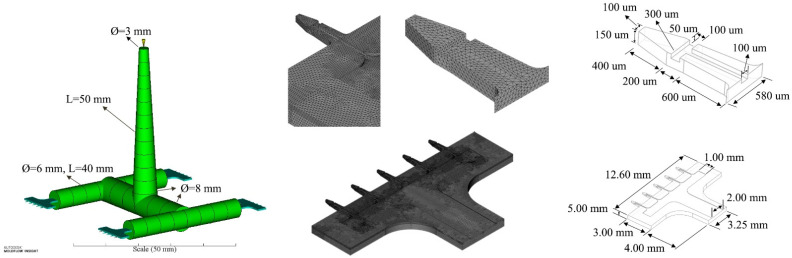
Finite element model, and the main dimensions of the microneedle array product.

**Figure 2 polymers-15-01199-f002:**
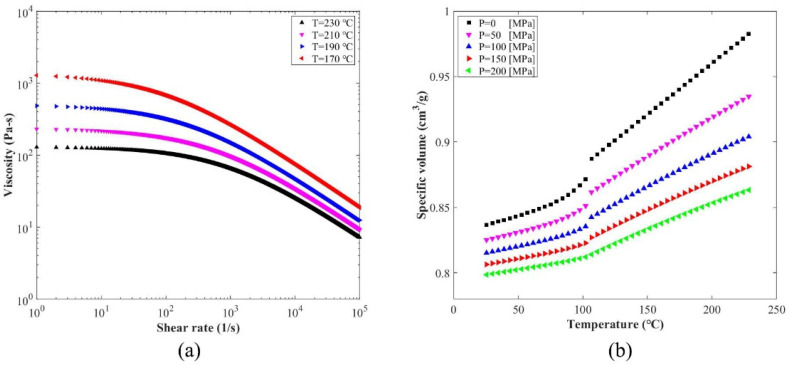
Viscosity and material state curves for PLA 3251D. (**a**) Cross-WLF viscosity model; (**b**) the modified 2-domain Tait pvT model.

**Figure 3 polymers-15-01199-f003:**
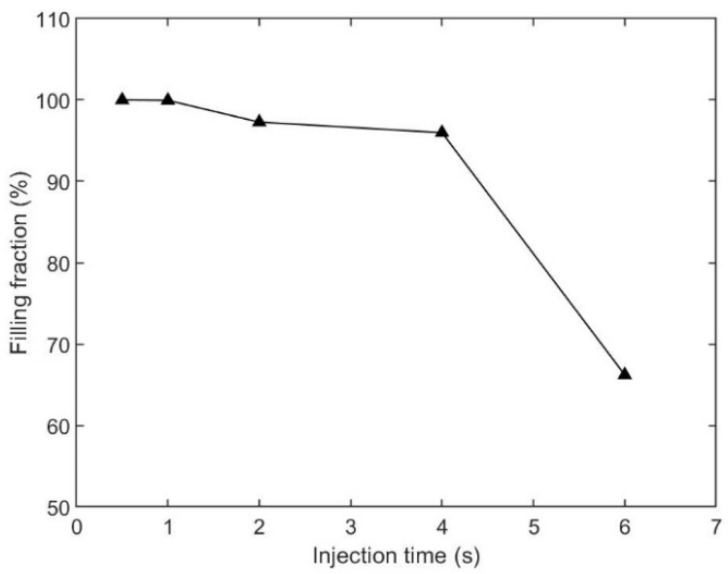
Relationship between the injection time and filling fraction, based on Table 3.

**Figure 4 polymers-15-01199-f004:**
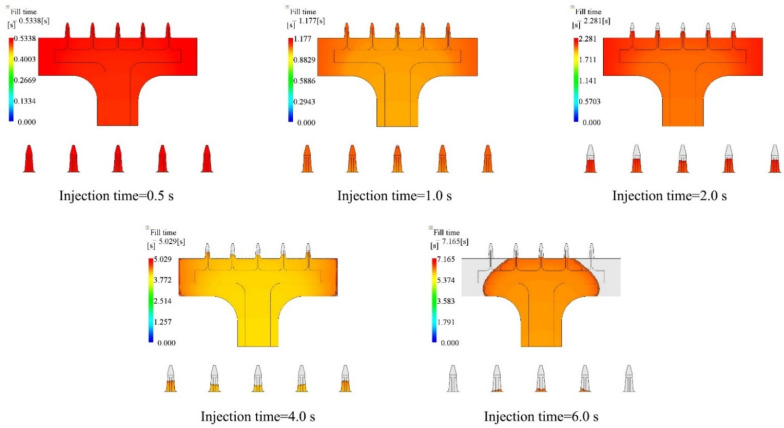
Filling contours for different injection time cases.

**Figure 5 polymers-15-01199-f005:**
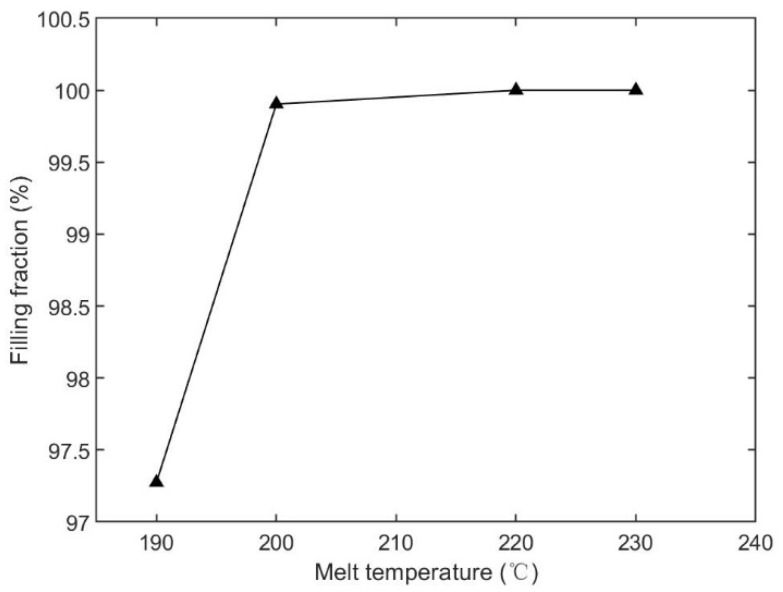
Relationship between the melt temperature and filling fraction, based on Table 4.

**Figure 6 polymers-15-01199-f006:**
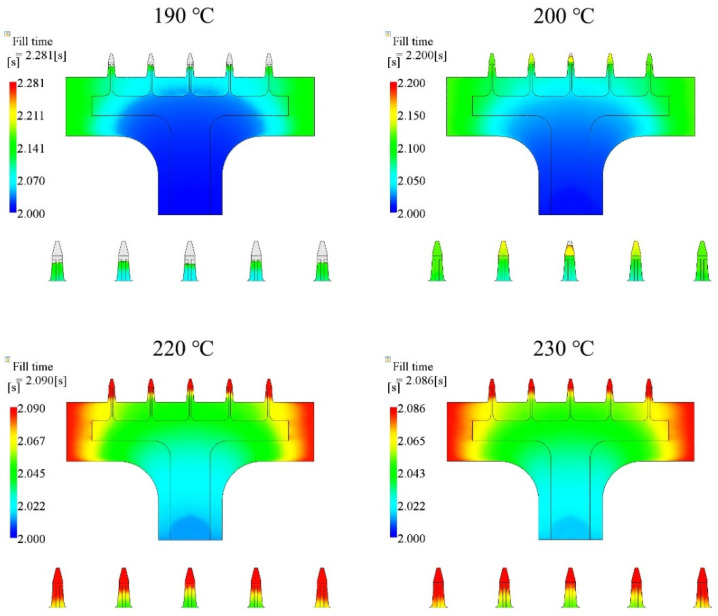
Filling contours for different melt temperature cases.

**Figure 7 polymers-15-01199-f007:**
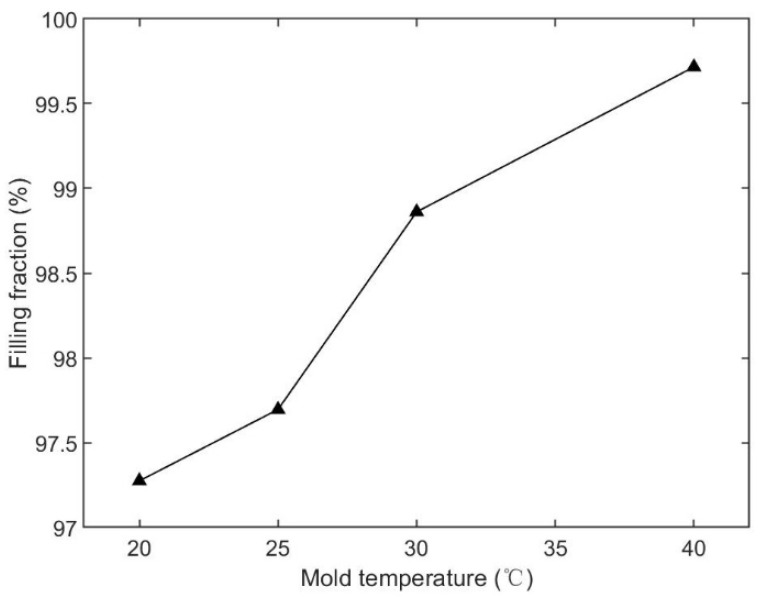
Relationship between the mold temperature and filling fraction, based on Table 5.

**Figure 8 polymers-15-01199-f008:**
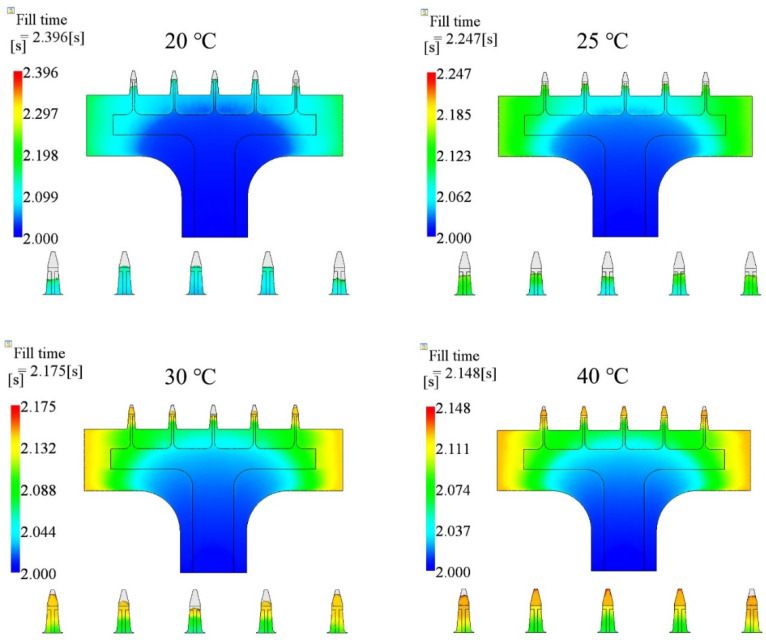
Filling contours for different mold temperature cases.

**Figure 9 polymers-15-01199-f009:**
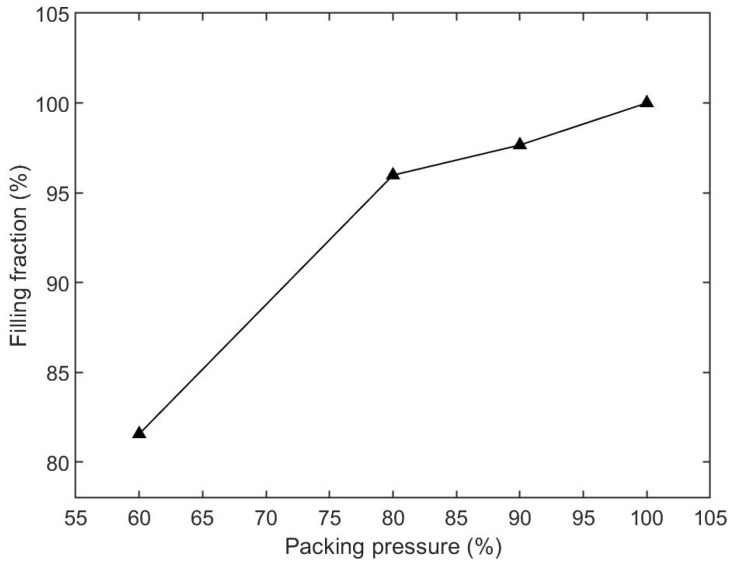
Relationship between the packing pressure and filling fraction, based on Table 6.

**Figure 10 polymers-15-01199-f010:**
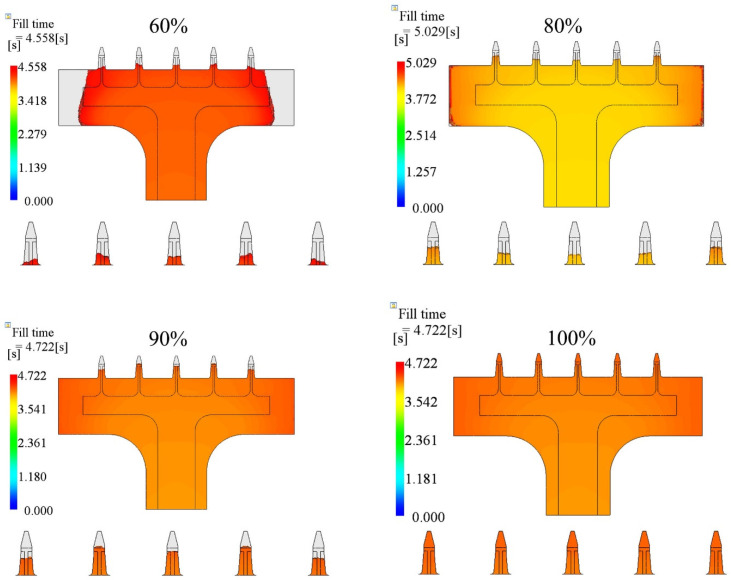
Filling contours for different packing pressure cases. The packing pressure was changed from 60% to 100% of the final filling pressure.

**Figure 11 polymers-15-01199-f011:**
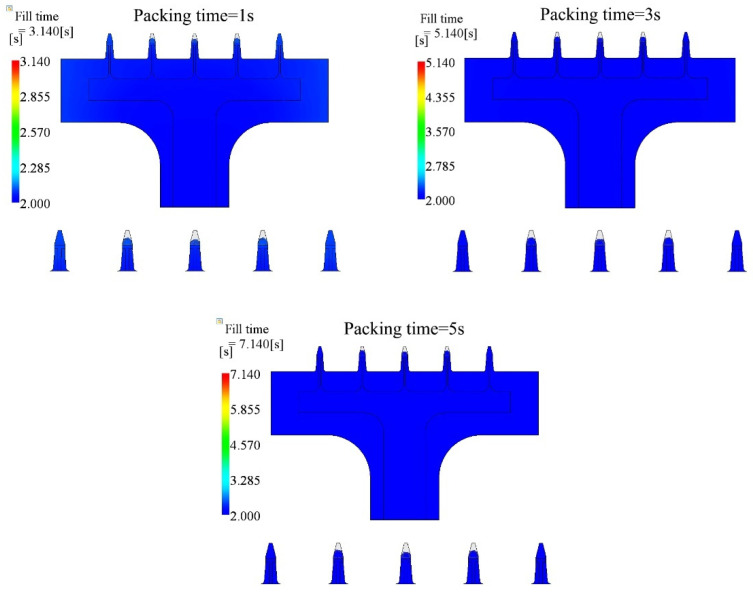
Filling contours for 1 s and 5 s packing time cases.

**Figure 12 polymers-15-01199-f012:**
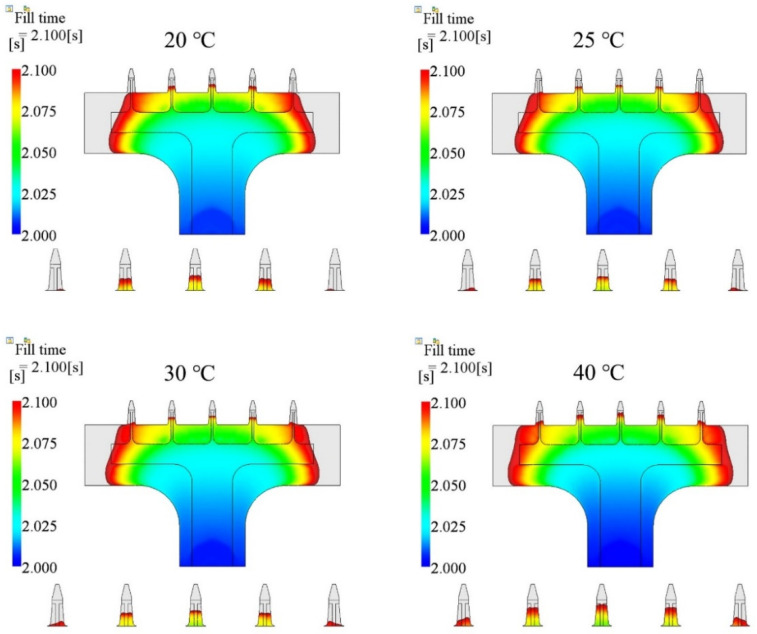
Filling contours using different mold temperatures at the same moment.

**Figure 13 polymers-15-01199-f013:**
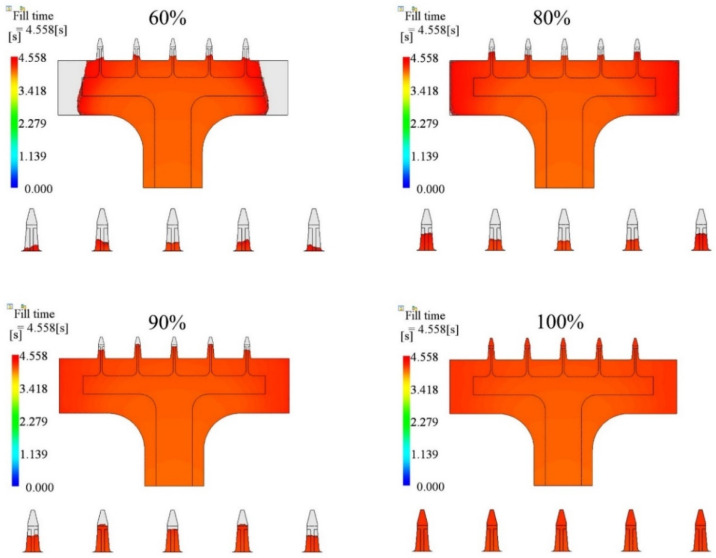
Filling contours using different packing pressures at the same moment.

**Figure 14 polymers-15-01199-f014:**
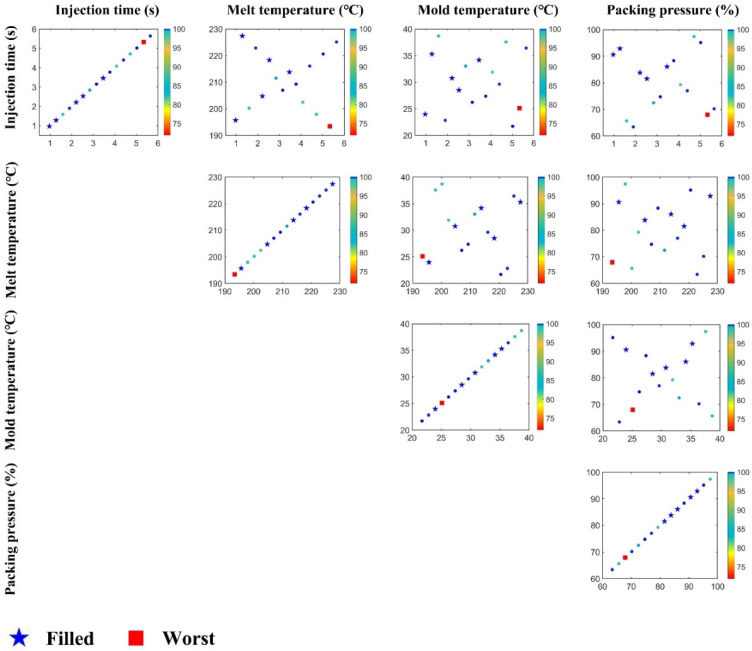
Combination plots of OLHS. Each subfigure is a scatter plot of two parameters. All filling fraction results of the OLHS samples are distributed in these two-parameter spaces.

**Figure 15 polymers-15-01199-f015:**
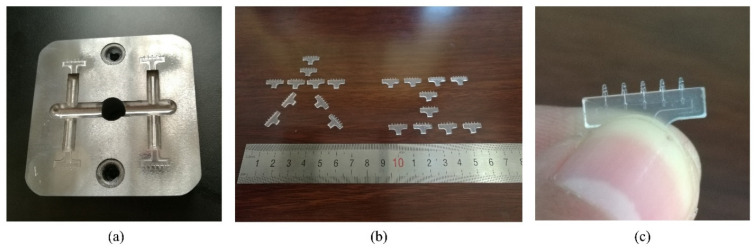
Actual mold and product. (**a**) Injection molding mold; (**b**) eventual product; (**c**) single microneedle array product.

**Table 1 polymers-15-01199-t001:** Cross-WLF viscosity model coefficients for PLA 3251D, in Equation (3).

Coefficients	*n*	*τ** [Pa]	$D_{1}\;[\mathrm{Pa}-s]$	$D_{2}\;[K]$	$D_{3}\;[K/\mathrm{Pa}]$	$A_{1}$	${\bar{A}}_{2}\;[K]$
Value	0.3846	1.29 × 10^5^	2.045 × 10^7^	373.15	0	16.71	51.6

**Table 2 polymers-15-01199-t002:** The modified 2-domain pvT model coefficients in Equation (7).

Coefficients	$b_{1m}$ $\left[m^{3}/kg\right]$	$b_{2m}$ $\left[m^{3}{\left(kg-k\right)}^{-1}\right]$	$b_{3m}$ [Pa]	$b_{4m}$ $\left[K^{-1}\right]$	$b_{1s}$ $\left[m^{3}kg^{-1}\right]$	$b_{2s}$ $\left[m^{3}{\left(kg-k\right)}^{-1}\right]\;$	$b_{3s}$ [Pa]	$b_{4s}$ $\left[K^{-1}\right]$	$b_{5}$ [K]	$b_{6}$ $\left[{\mathrm{KPa}}^{-1}\right]$	$b_{7}$ $\left[m^{3}kg^{-1}\right]$	$b_{8}$ $\left[K^{-1}\right]$	$b_{9}$ (1/Pa)
Value	8.936 × 10^−4^	7.831 × 10^−7^	1.268 × 10^8^	5.315 × 10^−3^	8.605 × 10^−4^	2.67 × 10^−7^	2.277 × 10^8^	3.16 × 10^−3^	388.15	5 × 10^−8^	3.276 × 10^−5^	6.353 × 10^−2^	9.922 × 10^−9^

**Table 3 polymers-15-01199-t003:** Processing parameters of the injection time study. Only injection time was changed.

Parameters	Injection Time [s]	Melt Temperature [°C]	Mold Temperature [°C]	Packing Pressure [%]	Packing Time [s]	Filling Fraction [%]
Value	0.5	190	25	80	10	100
	1.0					99.95
	2.0					97.27
	4.0					95.98
	6.0					66.21

**Table 4 polymers-15-01199-t004:** Processing parameters of the melt temperature study. Only melt temperature was changed.

Parameters	Injection Time [s]	Melt Temperature [°C]	Mold Temperature [°C]	Packing Pressure [%]	Packing Time [s]	Filling Fraction [%]
Value	2.0	190	25	80	10	97.27
		200				99.90
		220				100
		230				100

**Table 5 polymers-15-01199-t005:** Processing parameters of the mold temperature study. Only mold temperature was changed.

Parameters	Injection Time [s]	Melt Temperature [°C]	Mold Temperature [°C]	Packing Pressure [%]	Packing Time [s]	Filling Fraction [%]
Value	2.0	190	20	80	10	97.27
			25			97.70
			30			98.86
			40			99.72

**Table 6 polymers-15-01199-t006:** Processing parameters of the packing pressure study. Only packing pressure was changed.

Parameters	Injection Time [s]	Melt Temperature [°C]	Mold Temperature [°C]	Packing Pressure [%]	Packing Time [s]	Filling Fraction [%]
Value	2.0	190	25	60	10	81.57
				80		95.98
				90		97.66
				100		100

**Table 7 polymers-15-01199-t007:** Processing parameters of the packing time study.

Parameters	Injection Time [s]	Melt Temperature [°C]	Mold Temperature [°C]	Packing Pressure [%]	Packing Time [s]	Filling Fraction [%]
Value	2.0	190	20	100	1	99.39
					3	99.44
					5	99.44

**Table 8 polymers-15-01199-t008:** Parameters’ limits for OLHS.

Parameters	Injection Time [s]	Melt Temperature [°C]	Mold Temperature [°C]	Packing Pressure [%]
Lower	0.5	190	20	60
Upper	6.0	230	40	100

**Table 9 polymers-15-01199-t009:** Processing parameters of the packing pressure study, according to [33].

	Parameters	Injection Time [s]	Melt Temperature [°C]	Mold Temperature [°C]	Packing Pressure [%]	Filling Fraction [%]
ID						
1		3.46	213.80	34.17	86.07	100.00
2		2.84	211.53	33.03	72.47	99.53
3		2.21	204.73	30.77	83.80	100.00
4		2.53	218.33	28.50	81.53	100.00
5		1.59	200.20	38.70	65.67	98.94
6		1.90	222.87	22.83	63.40	99.88
7		0.97	195.67	23.97	90.60	100.00
8		1.28	227.40	35.30	92.87	100.00
9		3.15	207.00	26.23	74.73	99.99
10		3.77	209.27	27.37	88.33	99.90
11		4.40	216.07	29.63	77.00	99.79
12		4.08	202.47	31.90	79.27	98.03
13		5.02	220.60	21.70	95.13	99.91
14		4.71	197.93	37.57	97.40	99.23
15		5.64	225.13	36.43	70.20	99.80
16		5.33	193.40	25.10	67.93	71.87

## Data Availability

Not applicable.

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
