# Peer review of "Study on the Influence of Microinjection Molding Processing Parameters on Replication Quality of Polylactic Acid Microneedle Array Product"

_polymers, 2023, doi:10.3390/polym15051199_

Round 1

Reviewer 1 Report

This article focusses on the micro injection molding with the case of microneedle product. In general, the content is good enough for publishing. However, authors have to make clear these issues:

- In the introduction, authors should confirm that the replication quality is estimated by the filling ability, and this article does not cover the shrinkage or warpage.

- Line 17-20 (Abstract part): Actually, these results are almost the same with common cases of injection molding process, so, please show out the bright result (compare with common cases)

- Figure 1 has to be make clear (the lines are not clear).

- Please fill the data of meshing, especially at the micro feature area (element size, meshing type,…)

- In part 3, Table 1,2 and Figure 2 should have the reference if these data from other sources

- Table 3 should be presented in 1 page

- In this research, the mold temperature was varied from 20oC to 40oC. Authors should have some explanations. There are many papers mention about the influence of mold temperature on the filling ability, so, higher mold temperature, the more filling ability will be.

- Result in part 4.3: why didn’t you increase the mold temperature for reducing other parameters as pressure, melt temperature,…?

- Part 4.5: there is online 2 case of packing time. Authors should have some explanations.

- Line 33: In real case (experiment case), how did you control exactly the value of these parameter as in ID 7 (Table 7)?

- In experiment case, how did you define the filling is 100% or not?

- In Table 7, there are some cases, which have the filling is 100%, why did you select the ID7 for experiment ?

- In my opinion, the discussion could be inserted into the result part. However, this structure is still accepted.

Author Response

Thank you so much for all your constructive comments and questions. The answer for your comments and questions are as below:

This article focusses on the micro injection molding with the case of microneedle product. In general, the content is good enough for publishing. However, authors have to make clear these issues:

- In the introduction, authors should confirm that the replication quality is estimated by the filling ability, and this article does not cover the shrinkage or warpage.

Answer: Thank you so much! We revised the manuscript according to your comment. Actually, we reviewed many research work in the paragraphs from “Micro-injection molding (μIM) is…” to the paragraph “On the other hand,…” to state that filling is the important stage to improve replication quality for microinjection molded product.

- Line 17-20 (Abstract part): Actually, these results are almost the same with common cases of injection molding process, so, please show out the bright result (compare with common cases)

Answer: Thank you for your comment. Actually, we observed that the side microcavities could filled better than the central one under certain processing parameters. However, it doesn’t mean the side micro-cavities are filled better than the central one. The central micro-cavity was filled while the side micro-cavities were not under certain conditions in this study. We revised the manuscript accordingly.

- Figure 1 has to be make clear (the lines are not clear).

Answer: Thank you for your comment. We regenerated figure 1 in order to make it clear enough for publication.

- Please fill the data of meshing, especially at the micro feature area (element size, meshing type,…)

Answer: Thank you for your comment. There are two kinds of mesh used in this study. Beam element was employed for the whole runner system including the sprue. There are 34 beam elements and 35 nodes. 3D mesh of tetrahedron element was employed for the microneedle part. There are 2,231,732 3D tetrahedron elements with 5 element layers in thickness direction in this study to ensure the precise simulation about filling fraction. There are 468620 nodes in the mesh. We revised this part in the manuscript.

- In part 3, Table 1,2 and Figure 2 should have the reference if these data from other sources

Answer: Thank you for your comment. The parameters in table 1 and table 2 are from the database in Moldflow software. Figure 2 was plotted according to these parameters. All parameters were measured by Autodesk Moldflow company. We added this content into the revised manuscript.

- Table 3 should be presented in 1 page

Answer: Thank you for your comment. Table 3 was in the same page in our manuscript (word and pdf versions). We are not sure whether it was because the system issue. We try to solve this issue in the revised manuscript.

- In this research, the mold temperature was varied from 20oC to 40oC. Authors should have some explanations. There are many papers mention about the influence of mold temperature on the filling ability, so, higher mold temperature, the more filling ability will be.

Answer: Thank you for your comment. We agree that keep increasing mold temperature can improve the filling ability as you mentioned, especially for the product which has micro-features. The mold temperature range was used based on the processing parameter recommendation in Moldflow software database. We added the description about the parameter ranges in the revised manuscript to explain why we use these range, especially for mold temperature. The study in this manuscript was conducted in these ranges.

- Result in part 4.3: why didn’t you increase the mold temperature for reducing other parameters as pressure, melt temperature,…?

Answer: Thank you for your question. Yes, increase the mold temperature can reduce other parameters like pressure or melt temperature to obtain the similar results. This part was single factor investigation which was about the mold temperature. We chose four different mold temperature to show the influence about it “, and kept injection time (2.0s), melt temperature (190℃), packing pressure (80% of the final injection pressure) and packing time (10s) the same for all cases.” In other sections, it was the same, for instance, in section 4.2, we only changed melt temperature and keep parameters the same in all cases of section 4.2.

- Part 4.5: there is online 2 case of packing time. Authors should have some explanations.

Answer: Thank you for your comment. “Packing time should be also another parameter to improve the filling fraction since melts can be pushed into cavity under packing pressure. In this section, we also con-ducted the work about packing time. We chose two different packing time in the post-filling stage, 1s and 5s. The purpose was to show the effect on the filling fraction under different packing time.” However, we found that there was no apparent influence between 1s and 5s. So, we only performed these two cases in these sections.

- Line 33: In real case (experiment case), how did you control exactly the value of these parameter as in ID 7 (Table 7)?

Answer: Thank you so much for your question. We can’t guarantee the parameters are super exactly same as ID7, honestly. But we have precise mold temperature control system to reach the target value and ensure the lowest variation of mold temperature. We also have qualified plastification unit and injection unit which is made by German Boy company to ensure that the injection time, melt temperature and packing pressure reach the target values. Especially, for temperature, we started to inject material when the temperature reaches the values in ID7 to reduce the variation as much as we can.

- In experiment case, how did you define the filling is 100% or not?

Answer: Thank you for your question. This is super hard to define the filling is 100% or not. We focus on the volume filled by polymer, especially for the microfeature part. We used microscope to check whether the microfeature was filled entirely or not.

- In Table 7, there are some cases, which have the filling is 100%, why did you select the ID7 for experiment ?

Answer: Thank you so much for your comment. If we compare with other cases, which are ID1, ID3, ID4 and ID8, the injection time, melt temperature, mold temperature of ID7 were lower than other cases, except the packing pressure. Lower injection time means high fabrication efficiency, lower temperature means low energy consumption. Overall, it should be a good choice of low cost and high efficiency than others. We put this comment into the revised manuscript.

- In my opinion, the discussion could be inserted into the result part. However, this structure is still accepted.

Answer: Thank you for your comment. Actually, we planned to combine discussion part into results part. However, we thought different section should be clearer for read.

Reviewer 2 Report

The authors present a designed in-plane microneedle array product.

The correct use of the symbol micrometer in the document.

Given the response obtained based on the OLHS method, shown in table 7, they present five levels that reached a 100% filling state. Still, it is not clear to see the information in figure 14 where the correlation between the parameters should be observed.

Author Response

Thank you for your constructive comments and questions. The answers are as below:

The authors present a designed in-plane microneedle array product. The correct use of the symbol micrometer in the document. Given the response obtained based on the OLHS method, shown in table 7, they present five levels that reached a 100% filling state. Still, it is not clear to see the information in figure 14 where the correlation between the parameters should be observed.

Answer: Thank you for your comments. In this part, we tried to show the influence of injection time, melt temperature, mold temperature and packing pressure by using Orthogonal Latin Hypercube sampling (OLHS) method. We plotted these results in Figure 14. Figure 14 illustrated the scatter plot between any two parameters, like injection time and melt temperature, mold temperature and packing pressure, etc. The cases in which the cavity was filled entirely were marked with blue stars. So people can see the distribution in any 2 parameters space which can easily observe. It indicated that not any combination can reach 100% in another point of view.

Reviewer 3 Report

The manuscript is nicely written and presented. All images are captured well. 

I have some suggestion which will further improve the manuscript. 

Author should replace "um" with "µm". 

Authors should comment on the safety of prepared microneedle array product. 

Author should also comment on the skin irritation potential of microneedle array product. 

The figure legends and Tables caption need improvement. All legends and captions should have enough description for a reader to understand the figure/table without having to refer back to the main text of the manuscript.

Authors should at the minimum proof read the entire manuscript for typographical errors and fix all grammatical errors.

Value and unit should be separated by a space e.g. 20 ± 2 g (except for % and °C “degrees Celsius”).

Author Response

Thank you so much for your constructive comments and questions. The answers are as below:

The manuscript is nicely written and presented. All images are captured well. I have some suggestion which will further improve the manuscript. 

Author should replace "um" with "µm". 

Answer: Thank you for your suggestion. The unit of "um" was replaced with "µm" accordingly in the revised manuscript.

Authors should comment on the safety of prepared microneedle array product. Author should also comment on the skin irritation potential of microneedle array product. 

Answer: Thank you for this comment. This is very nice comment about this product. Actually, skin irritability issue depends on people’s tolerance about PLA material. PLA is one of famous biodegradable polymer materials which is proved to be very nice biocompatibility for human. We have already added the comment into the revised manuscript to describe about this part.

The figure legends and Tables caption need improvement. All legends and captions should have enough description for a reader to understand the figure/table without having to refer back to the main text of the manuscript.

Answer: Thank you so much. We have revised accordingly.

Authors should at the minimum proof read the entire manuscript for typographical errors and fix all grammatical errors.

 Answer: Thank you so much. We have revised the entire manuscript to fix the grammatical errors. We hope the revised version meet the requirement for publication in this journal.

Value and unit should be separated by a space e.g. 20 ± 2 g (except for % and °C “degrees Celsius”).

 Answer: Very appreciated! We also notice this problem in our manuscript. We have read through the entire manuscript and revised accordingly.

Round 2

Reviewer 1 Report

Dear Authors,

I would like to commend you on the improvements made in your new version. It is much clearer and more understandable for readers. However, I have two minor comments on your last version:

Regarding the issue of "there is only 2 cases of packing time. Authors should have some explanations," if possible, it would be beneficial if you could add the case of 3 seconds to prove the result that "there was no apparent influence between 1 second and 5 seconds."

With regard to the defining of 100% filling or not, it would be helpful if you could add an explanation into your manuscript to clarify this point.

Thank you for considering my comments. I look forward to seeing the online version of your manuscript.

 Best regards,

Author Response

We appreciated that you gave us constructive comments to improve the manuscript for publication. Please see the answers below:

  1. Regarding the issue of "there is only 2 cases of packing time. Authors should have some explanations," if possible, it would be beneficial if you could add the case of 3 seconds to prove the result that "there was no apparent influence between 1 second and 5 seconds."

Answer: Thank you for your comment. We added another case that the packing time is 3 second according to your comment. It was added in figure 11.

  1. With regard to the defining of 100% filling or not, it would be helpful if you could add an explanation into your manuscript to clarify this point.

Answer: Thank you for your comment. We added the explanation in the latest version according to your comment. We agree that it should be more helpful for readers. We put it into the paragraph ‘If we compare with other cases, which are ID1,…’.
